# Strengthening Food Security Near the Arctic Circle: Case Study of Fairbanks North Star Borough, Alaska

**Robert W. Orttung** [1,*]**, James Powell** [2]**, James Fox** [3] **and Claire Franco** [4]

1   Sustainable GW and Elliott School of International Affairs, The George Washington University, Washington, DC 20052, USA

2   Master of Public Administration Program, School of Arts and Sciences, University of Alaska, Southeast, Juneau, AK 99801, USA; jim23powell@gmail.com

3   Fairbanks North Star Borough Sustainability Commission; Fairbanks, AK 99701, USA; Jimmy.Fox@fnsb-bc.us

4   Department of Geography and Elliott School of International Affairs, The George Washington University, Washington, DC 20052, USA; claire_franco@gwmail.gwu.edu

*   Correspondence: rorttung@gwu.edu; Tel.: +1-202-994-5730

**Abstract:** Reliable food supply is a central concern for residents of cities located in remote locations with extreme climate conditions. The purpose of this article is to examine how stake-holders in such northern cities ensure a high level of food security. We examine a case study of the Fairbanks North Star Borough, Alaska, which is located in the interior of the state near the Arctic Circle. Borough policymakers are seeking to address community concerns through a collaborative, multi-stakeholder process of working with local farmers, distributors, consumers, activists, and academics. We examine the effectiveness of this process through participant-observation and process tracing of the initial results of the newly established Fairbanks North Star Borough sustainability commission. The new commission has adopted a sustainability plan drawing upon the input of community stakeholders, but it remains to be seen how the plan will be implemented and if it will meet the needs of diverse groups within the community. This analysis makes a contribution by examining the hypothesis that university-based teams and public input can improve public policy outputs in the area of food security by organizing their work around a focus on data. Specifically, the article examines the most effective mechanisms for collaboration among academics and policymakers to incorporate public input into food security policies.

**Keywords:** food security; Arctic; community engagement

## 1. Introduction

Reliable food supply is a central concern for remote communities located in extreme climate conditions. How are these communities working to marshal their resources to ensure that all citizens have access to sufficient food supplies? This article develops a case study of the Fairbanks North Star Borough, Alaska, USA, to address this issue. It examines how data serves as an organizing principle with policymakers, university-based academics, and the general public so they can coordinate and produce effective public policy.

Given the extreme climate, securing food has been a primary concern for humans in interior Alaska well before recorded history. Approximately 6000 years ago, the Athabascan peoples arrived [1]. With the absence of written records, it is difficult to determine to what extent humans lived sustainably in this part of the Arctic. Wood bison was a likely source of food in the prehistoric era and some oral history reports from elders suggest that early inhabitants of the Yukon Flats may have overhunted them during periods of starvation, eliminating them from Alaska [2]. Other evidence suggests that the

bison disappeared before humans entered the area [3]. Unfortunately, the currently available evidence is not strong enough to draw a definitive conclusion. Athabascans reported no major shifts in food sources until approximately 100 years ago, with moose hunting emerging as a relatively new tradition to groups in the region [1]. Since colonization and settlement of nonindigenous people and the recent rapid climate change impacts, Athabascan subsistence systems may change again, providing valuable lessons for all residents of the region.

More recently, we know that the Fairbanks region (hereafter "Fairbanks"), and particularly the Matanuska Valley, had hundreds of small farms just 100 years ago [4]. Nevertheless, from Fairbanks' beginnings at the dawn of the twentieth century, residents relied on food imported from outside the state. In 1904, when Fairbanks was a gold mining town, there were few local vegetables, and miners and their families consumed canned food shipped from thousands of miles away. Staples included canned milk, tomatoes, peas, and corn, as well as dried fruit, flour, sugar, beans, split peas, and cereals [5]. People bought or grew their own food in the summer and preserved it during the winter in their root cellars, which stored potatoes, turnips, carrots, celery, and other garden items for months. As late as 1955, Alaskans produced roughly half of their food.

Today, local food production makes up less of local diets than ever in the past. Economic globalization, including advances in the ease of transportation, have dramatically changed the lifestyle of people living in the far north, in part by making it much cheaper and easier to import food from the outside [6]. As a result, Alaskans currently obtain as much as 98 percent of their food supply from out of state [7]. In this context, the global food chains that make it possible to eat easily, and relatively cheaply, food from distant places also make it possible to escape the need to rely on local production [8,9].

Currently, food is a $5 billion a year business in Alaska [10]. Typically, Alaskan products, such as salmon, are shipped to outside customers by firms whose owners do not live in Alaska [10]. Simultaneously, Alaskans spend $1.9 billion a year to import food [10]. "Essential items arrive by airplane, barge, and truck from Mexico, Europe, Asia, and the continental U.S., while much of Alaska's maritime bounty is channeled to Asia", according to a 2018 report [10]. While this system generally provides a reliable supply of reasonably priced food, its major drawback is that it relies heavily on long supply chains, which lead to extensive fossil fuel use to move the goods with the resulting negative impacts on the environment. These supply chains are also vulnerable to natural disasters and Alaskans worry about what they would eat if suddenly they were cut off from distant food sources. Figure 1 shows the current food system with its heavy reliance on food from outside and relatively little local production. To the extent that there is local food production, it is more for individual consumption and not hooked in the broader food system. The problems are even more acute outside urbanized areas [11].

This article develops a case study of the Fairbanks North Star Borough (FNSB) and its population hub Fairbanks, a network of neighborhoods in the southern portion of the FNSB. The case study addresses the bigger question of how policymakers, academics, and the general public can develop strategies for managing food security challenges through a focus on data. With a population of roughly 100,000, the FNSB serves as a hub for Alaska's interior and northern regions. Although Fairbanks started as a goldmining town, today the flagship campus of the University of Alaska and the two military bases—Wainwright (army) and Eielson (air force)—shape its character [5]. While we believe FNSB as a case study can shed considerable light on the difficulties faced by the Arctic and subarctic cities in general, it is important to keep in mind the enormous variation across the Arctic. Of course, the similarities between Fairbanks and other northern cities only goes so far. Fairbanks' high cost burden is similar to other communities in Alaska (Census Bureau Cost of Living Index—Selected Urban Areas—Census Bureau https://www2.census.gov/library/publications/2011/compendia/statab/.../12s0728.xls (accessed on 14 April 2019)). Fairbanks may be similar to Rovaniemi, Finland in its urban sprawl, for example, but across the circumpolar north there is tremendous variation in city size, access to food supplies, and patterns of governance [12].

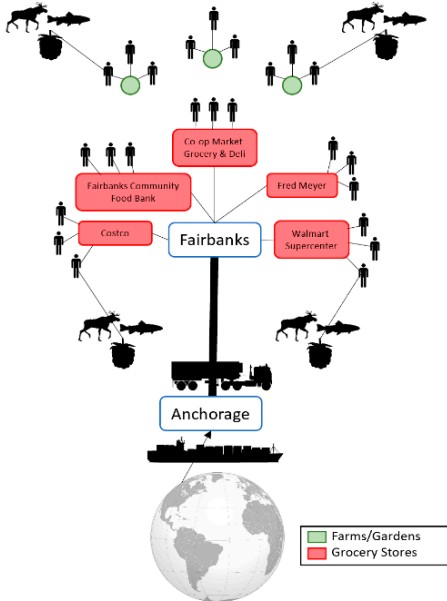

**Figure 1.** Existing Fairbanks North Star Borough Food System.

While Fairbanks' location and climate may make it an extreme case, reliance on imports for food security affects at least 1.3 billion people around the world [13]. These people live in locations where current food production is not sufficient to feed all residents. Importing food is considered the lowest cost strategy to provide food to the population, with dependence on outside sources seen as a tolerable risk.

Despite the problems with the current system, it would not be a simple matter to increase local food production in the FNSB. Importing farm inputs is expensive and often means that farmers in the continental U.S. can produce crops at a fraction of the cost required of Alaskan farmers. Once-viable Alaskan cattle ranches and dairies disappeared to make way for more lucrative subdivisions to house incoming residents [14]. The loss of the earlier farms means that young Alaskans now have little knowledge of food production and there is little support for this kind of activity in cities like Fairbanks. Currently, Alaskan farms produce about $15 million a year in food products [10]. Wild food, such as fish and meat caught by Alaskans, has an estimated value of between $400 and $900 million per year [10]. Small farms sell about $2.2 million in products to local customers throughout the state.

The Alaska State House of Representatives set up a food security subcommittee on 2 May 2018. Similarly, in 2018, the newly-established Fairbanks North Star Borough sustainability commission identified food security, along with energy and solid waste, as one of the top three priorities for the community, and in January 2019 adopted a plan laying out priorities and benchmark metrics to address the issue (The plan is available at http://fnsb.us/Boards/Pages/SustainabilityCommission.aspx (accessed on 5 April 2019)). Residents of Fairbanks believed that the main problem they faced in the realm of food was that they imported nearly all of their food from beyond state boundaries. Local citizens worried that if supply lines were cut as the result of an extraordinary hazard, they would face severe shortages because there was a widely-held belief that Fairbanks only had a 3-day supply of food on hand.

The central hypothesis at the core of this case study research effort is the claim that public input and expert advice make it possible to generate more effective public policy and that a focus on data is the best way to facilitate collaboration among policymakers, experts, and the public. The aim of the analysis is to trace the process by which policymakers, university academics, and the public coordinated in this case, and to examine the costs and benefits of the process. While this article can only make limited claims because it focuses on one city and describes a process still underway, it seeks to show what worked in this case and what might potentially work in other cities as well.

The rest of the article proceeds in the following way. First, it situates this case study in the existing literature. Second, it lays out the participant-observation, interview, and process tracing methodologies. Then, it describes the evidence from the case study. The discussion and conclusion summarize what worked in the FNSB and what lessons can be drawn for other cities.

*Literature Review*

The Fairbanks case study allows us to build on the existing literature across a range of topics, including:

- the effectiveness of collaboration between policymakers and academics in pushing forward public policies,
- the effectiveness of public input in policy making,
- and the role of public discussion in addressing the needs of all elements of the community.

First, the importance of collaboration between university-based academics and policymakers in addressing complicated policy problems is increasingly a focus of attention. Universities and the corporations and research institutions they spin off are frequently cited in spurring economic growth; Stanford University for sparking the emergence of Silicon Valley, and Carnegie Mellon for helping Pittsburgh to transition from a rust belt city to a center of robotics innovation. But, beyond such direct economic impacts, universities can help stimulate innovative policy making. For example, universities can facilitate a new kind of networked problem-solving that brings together public officials, scientists, corporate leaders, and community activists. Just as universities can serve as economic engines, they can stimulate public policy solutions to pressing problems. Universities provide a neutral ground where stakeholders can meet and work out solutions while taking advantage of the knowledge of the faculty. In this context, the university as an institution serves as an "honest broker" [15]. The key function of the honest broker in providing policy advice is helping to create new options to overcome gridlock in situations where other options are blocked.

The rapid proliferation of new data sources is pushing forward the boundaries of science, but policymakers often lack the ability to translate the new knowledge into effective solutions that positively impact their constituents. Food security, the focus of our case study, has characteristics of a "wicked problem" because it crosses multiple governmental domains including transportation, environment, and health [16]. In dealing with complicated problems that affect numerous public policy areas, research and practical experience show that multi-disciplinary teams of scientists [17] must find ways to work with city officials, entrepreneurs, citizen groups, and other stakeholders [18]. Such efforts place heavy burdens on participants in terms of communicating with each other and coordinating their work. These efforts can pay off since working with data provides a common ground that can satisfy the interests of both policymakers and academics [19].

Second, the case study gives some insight into the value of public input into the policy-making process. While some observers of this process argue that citizen input through the process of public commenting has little impact on policies adopted [20], other studies show that city leaders are in fact broadly responsive to public concerns [21]. Moreover, there is growing evidence that members of the public want to participate in the policy-making processes and are capable of high quality deliberation, especially when such deliberation is well arranged, including the "provision of balanced information, expert testimony, and oversight by a facilitator" [22]. While soliciting public input for sustainability planning is typical in all geographical settings, it is considered to be particularly important in the Arctic [6]. As elsewhere, policies that do not have public input from the start may have little support from the population and therefore may be less likely to be effective in their implementation.

Finally, to what extent does public participation in policy making lead to just outcomes that address the needs of all citizens? It is commonplace to argue that public participation is crucial to effective policy-making, but does it produce just outcomes? Evidence drawn from analyzing who speaks at planning and zoning board meetings, shows that participants tend to be older, male, longtime

residents, voters in local elections, and homeowners [23]. In other words, engagement is a luxury for those who are well off. Likewise, Fainstein argues that mere participation in decision-making is not enough to ensure just outcomes and that the process has to include considerations of diversity and equity as well [24].

Building on these works, we seek to test whether data can serve as a way to bring together a variety of stakeholders—policymakers, experts, and members of the interested public. We focus on the FNSB sustainability commission as the relevant arm of the borough government. We also seek to draw attention to understanding who participates in the process and ensuring that the outcomes meet the needs of the entire community, not merely its more vocal or well-connected members.

## 2. Materials and Methods

This work draws on the experience of the authors as participants and observers in contributing to the FNSB sustainability plan and these affiliations are listed explicitly as conflicts of interest that affect the nature of the analysis presented in this article. James Fox is a member of the FNSB sustainability commission and spearheaded the process of developing and writing the plan. James Powell and Robert Orttung served as advisors to the commission. In this capacity, they conducted 21 interviews with stakeholders in Fairbanks during 22–29 July 2018 under the aegis of the George Washington University Institutional Review Board. These stakeholders included:

- six of the seven commissioners of the FNSB sustainability commission,
- liaisons from the sustainability commission to key groups in the community, including the military bases, the faith community, the University of Alaska, Fairbanks, and educators,
- farmers working in the area,
- the head of the local foodbank,
- young Vista program activists working on food security issues,
- members of the Mayor's office,
- and officials in the related fields of energy and waste management.

Each interview was conducted according to a semi-structured list of questions [25]. These questions are listed in Appendix A. The interviewers started the conversation with each respondent by asking what they thought were the key issues facing the community in general. Then they were asked about the most pressing problems in the area of food security. Having established the most important issues, the questions turned to what each respondent thought were plausible solutions, who would be the most important actor to implement these solutions and what resources they would need. The questions specifically asked about the role of policymakers, the private sector, and members of the community, particularly young people. Finally, the interviewers asked what kind of resources would be necessary from outside the community.

Orttung and Powell listened to the meetings of the sustainability commission over the phone and provided input where their expertise was appropriate. Since they interviewed all but one member of the sustainability commission, the interview process gave the academics and policymakers a chance to engage in conversation and develop a common understanding of the problem and how best to start a conversation with the community about it. The academics also played a role in collecting community input and summarizing it into possible action items. These reports made it easier for the policymakers to sift through the various recommendations from the public input and determine which ones could most usefully be employed.

Claire Franco, a student at George Washington University, took the lead in developing the visual dashboards that will be used to communicate the key elements of the sustainability plan to members of the general public. These dashboards, discussed in greater detail in Section 3.5, serve as a way of encouraging greater community input into the process of setting metrics and goals. The sustainability plan identified three priority areas for the FNSB—food security (the focus of this article), energy, and solid waste management—but it did not set any specific goals in the food or energy areas.

The dashboards seek to encourage an engaged public to set these goals and help to identify ways to achieve them.

The study also employs a process-tracing approach in order to track the progression that the sustainability commission used to develop and adopt its sustainability plan [26]. We use process tracing as a central method for determining the causal mechanisms linking university-based expert and general public input into policy making processes and the production of public policy outputs. Interviews with the key players involved provide one of the most useful sources of data for describing these causal mechanisms [27].

## 3. Results

This section lays out the heart of the case study. It starts with the establishment of the FNSB sustainability commission, then examines stakeholder input into the development of the sustainability plan, general public input, adopting the plan, and the first steps in implementing the plan.

### 3.1. Institutional Innovation

On 12 October 2017, the Fairbanks North Star Borough Assembly established a new sustainability commission. The resolution establishing the new body cited research from other communities around the world that had demonstrated that such institutions could save money, reduce municipal operating expenses, stimulate business, improve human wellbeing for current and future generations, promote more holistic planning, and enhance food security [28]. The FNSB sustainability commission was built on the basis of the municipality's existing borough recycling commission. Many of the duties that necessitated the authorization of the recycling commission with large public support in 2009 were completed with the establishment of the borough's central recycling facility on 1 September 2017, providing an opportunity to broaden the duties of the recycling commission into the sustainability commission.

The ordinance established a body of seven members (appointed by the mayor and confirmed by the borough assembly) who meet at least quarterly to discuss and update policies and projects under their responsibility. The committee's mandate includes reducing individual and collective ecological impacts while improving the economic, security, and sustainability of the borough.

As part of the expansion of its responsibilities, the sustainability commission inherited the duties of the agricultural commission. The new emphasis on policies relevant to local agriculture helped the new sustainability commission target the issue of food security. The purpose of moving from a commission devoted to recycling to one addressing sustainability was to increase the ability of the body to work in holistic ways. In addition to examining issues of food security and continuing to oversee solid waste management and recycling efforts in the borough, the sustainability commission explores avenues of local, renewable energy in order to help reduce supply volatility and emissions contributing to climate change, while also seeking to diversify the regional economy. In practice, the commission has the job of setting sustainability goals and making long range recommendations, policies, and budgets to realize them. The ordinance acknowledges the larger global drive toward sustainability while focusing attention on the details of the region's own population and practical problems. The integration of the former agricultural commission into the sustainability commission in this restructuring recognized the critical component of responsible food provision in the long-term implementation of any municipal planning. The sustainability commission serves in an advisory capacity and must seek to ensure that the mayor and borough assembly implement its plans and that the general public is generally aware of them and supportive of what the commission seeks to do.

### 3.2. Local Stakeholder Input

The 21 interviews that Powell and Orttung conducted for the sustainability commission identified the concerns listed in Table 1. The top concern among stakeholders was food supply and the vulnerability of the transportation system to disruption. Given Alaska's and Fairbank's reliance on

out-of-state supplies for nearly all of its food, stakeholders worried that big earthquakes or volcanic eruptions could limit in-bound flights and other forms of transportation. The potential loss of air service was particularly important since Fairbanks is the end of the railroad line and is therefore the last place to be serviced. There was a common perception that, several years ago, grocery stores had several weeks of food in stock. Now, with a greater focus on just-in-time delivery techniques, the perception is that there are only a few days of food supply on hand. Given Fairbanks' remote location and cold climate, there is concern that such a business model might not be appropriate for the region.

**Table 1.** Concerns about Food Security Identified by Fairbanks Stakeholders.

| Concern | Issue |
|---|---|
| Food supply | • Big earthquake, volcanic eruption can cut down air travel<br>• Fairbanks is the end of the line for the road, railroad network<br>• Several years ago, grocery stores had several weeks of food. Now it is only a few days in storage. This business model might not work for a place like Fairbanks. |
| Infrastructure | • Lack of cold storage<br>• Hard to create agricultural infrastructure when growing season is short<br>• Energy costs are high, so hard to do indoor farming<br>• Land is publicly owned with limited access<br>• Roads are for mineral resources and military, food is an afterthought<br>• Lack of food processing facilities in Fairbanks area |
| Labor | • Farmers say hiring affordable, qualified labor is difficult.<br>• No migrant workers |
| Accessibility | • Food prices are higher than elsewhere<br>• USDA declared South Fairbanks a food desert |

The stakeholders also identified a number of problems with infrastructure that make it difficult to ensure steady supplies of food in the region. Fairbanks lacks cold storage facilities that can ensure a reliable source of distribution during all seasons of the year, particularly the short but intense growing season during the polar summer. Since such facilities would only be usable for short periods of the year, it is much harder to justify financing for them. High energy costs in the far north naturally raise the price of inputs for all forms of agriculture, but particularly make it hard to develop indoor farming, which frequently is energy intensive. There is also a lack of food processing facilities in the Fairbanks area. A further problem is that the majority of arable land is publicly-owned and is not always readily available to farmers who would like to exploit it because of a lack of passable, all-weather roads open to the public.

Hiring appropriate labor is a major hindrance for many agricultural enterprises. There are few qualified workers on the market and often the cost of employing them is greater than what the farmers can afford. Similarly, there are few migrant laborers willing to work on such enterprises. Travelers from Mexico prefer to work on more lucrative projects, such as resource exploitation, or opening up their own restaurants [29].

Finally, accessibility to food for low-income residents can be difficult. Generally, given the long supply lines, food prices in Fairbanks are higher than they are elsewhere in the U.S. Similarly, the urban sprawl of the city means that residents who do not own their own vehicles might have difficulty accessing nutritious food supplies. The USDA has declared South Fairbanks, where relatively poor

members of the population are concentrated, a food desert. As Figure 2 shows, there are relatively few food assets in this part of town.

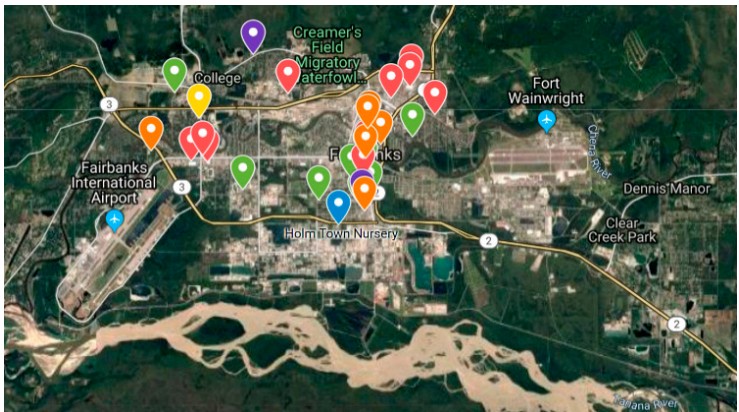

**Figure 2.** Map Showing Concentration of Food Assets in Fairbanks and Desert in South Fairbanks
Source: https://www.google.com/maps/d/u/0/viewer?mid=1Z3SkTJX4qPlgUz2--6CLx9fLMO_a4_Xo&
ll=64.82500344397585%2C-147.74696516319247&z=11.

On the basis of the interviews, the Fairbanks sustainability commission developed a draft set of indicators for food security as a way of launching a discussion with the broader public in order to define the most pressing concerns.

*3.3. General Public Input*

The indicators were: increase agricultural workforce development, increase number of days of supply of available food, increase local food production, and increase access to local fruits and vegetables. On 4 October 2018, the commission submitted these goals for discussion at a community open house at the Noel Wien public library in Fairbanks. The feedback from the public gave a strong sense that the work on the sustainability plan was on target. There was general agreement that the goals identified were important to the community, though some expressed a desire to expand them to take on bigger issues, such as climate change. Also, the feedback seemed to support the original rationale of picking the three goals in the first place. As one member of the public pointed out: "There is also a lot known about each of these topics so they are 'low-hanging fruit' where the sustainability commission can demonstrate strong early successes in its efforts." The rest of the general input focused on what kind of baseline data would be possible to collect, what the best metrics were to measure success going forward, what goals to set for the community to achieve, and how to define a strategy to best achieve those goals.

Some citizens appreciated the role that the sustainability commission played in taking the lead. One participant pointed out, "This summit got a lot of folks in the same room to share great ideas. I believe we came up with some good goals and identified barriers to those goals." However, there was one comment that was not supportive of the sustainability commission's efforts and particularly the role of the local government, in this case the FNSB, writing: "I am concerned that you are lumping $PM_{2.5}$ with trash, with food, with energy. What basically you appear to be doing, is a backdoor 'let's control everything' theory. The people that arrived here prior insanity (FNSB) struggled, but survived without starving to death, or freezing to death, (unless they were inebriated, which still occurs.) Who decided FNSB (who doesn't do such a great job at maintaining anything) be allowed to give any input to food security? It appears to be just 'a reason to have more employees.' There's my public input. For what it is worth, I feel that if people want to join together for anything sustainable, they should do it, and not have FNSB have their hands in it. You guys are relieved of holding my hand, thanks."

One participant expressed concern about the nature of the process, stating "It almost feels rushed regardless of the topics being on the agenda since the beginning of the year. I am curious to know more from the 'expert' analysts on these subjects and why those goals and indicators were suggested. If you had addressed these topics, it still feels like some have more questions than answers which at this point probably shouldn't be." Others suggested making sure that there would be interlinkages among the three topics (and others) chosen as the sustainability commission's priorities: "Consider how food waste can be utilized into compost and possibly even alternative energy practices."

In terms of food security content, the public input focused on several topics. The need to change the food culture in Fairbanks raised concern. Particularly, members of the public stressed the lack of a strong food culture in Fairbanks in which citizens actively participated in growing food locally, sharing, and eating among friends and family. Additionally, they lamented the lack of an ecosystem that provided strong linkages among farmers and customers. While these problems are typical for many locations, they have particular relevance in the Arctic, where there is a strong emphasis on self-reliance and a long tradition of neighbors voluntarily helping each other to cope with the challenges of living in an extreme climate.

One individual suggested increasing the amount of food that retail grocery stores hold in storage, potentially through the use of tax incentives. Nevertheless, overall, the focus was on individuals or the private sector taking on this task rather than making it a governmental responsibility.

Another motivation of the comments was ways to increase food production through a variety of mechanisms, including expanded community gardens on public land, school district property, and in private yards. The growing season could be extended with the use of solar-powered greenhouses. One citizen suggested adopting an idea from Palmer—placing public garden boxes for growing food in well trafficked areas which would bring greater awareness of food production to pedestrians and visitors to public spaces. Other ideas included solar-powered hydroponic farms, increasing the percent of local food sold by Fairbanks' grocery stores, setting up local canneries in order to can locally grown vegetables, and a community-run chicken farm. The public also called for increased educational resources devoted to increasing citizens' knowledge of how to grow their own food most effectively.

*3.4. Adopting the Plan*

Drawing on the expert summaries of the stakeholders' input and the feedback from the public, the sustainability commission adopted the plan at its January 2019 meeting. One last minute issue revolved around how much emphasis to place on questions of nutrition. Although poor eating habits and obesity are a problem in Fairbanks, as they are in most U.S. cities, several of the commissioners wanted to focus the plan specifically on the need to ensure food supplies in the face of hazards, arguing that preparedness is the real issue, rather than including much discussion of nutrition. These commissioners emphasized repeatedly in two sustainability commission meetings that most of the population was really concerned about availability and that issues of nutrition were secondary. Or, to be more precise, the issues were sequential, with questions of food supply of immediate concern and discussions of nutritional quality a longer-term issue.

In particular, the idea there were only three days of food in the city, whether accurate or not, animated discussion and focused the attention of both citizens and policymakers [7]. This prospect even summoned up visions of Fairbanks becoming something like the "thunderdome", the arena for jousting in the post-apocalyptic film Mad Max.

While the sustainability plan identified four key areas where it sought to push forward—increased agricultural workforce, food stocks, local food production and sales, and access to local fruits and vegetables—it did not identify key quantitative goals for the community to achieve in these areas. Rather, having identified the issues where community stakeholders and policymakers believed progress was needed, the plan sought to engage the community in a discussion about what the concrete goals should be. Launching this kind of discussion seeks to encourage community members to consider where they want to go and how best they think the borough can get there.

### 3.5. Implementing the Plan

Once the plan was successfully adopted, the crucial step was its implementation. The key issue facing state and local governments in Alaska is lack of revenue. The vast majority of income for government operations in Alaska comes from the federal budget. The state levies no personal income or sales tax, which are typical sources of revenue for U.S. states. With lower oil prices and lower production in recent years, state tax revenue has been dropping, forcing state and local leaders to make extensive cuts in their budgets. This general cost-cutting naturally makes officials reticent to take on additional projects.

The cost implications from the food security elements are not clear. Certainly, there is public concern about food security and support for making the food supply more resilient to environmental hazards, but it remains unclear how this general support will translate into backing for specific efforts to take action along the lines proposed in the plan. The next step is encouraging more community input to determine specific metrics and goals for the key areas that have been identified as important in the area of food security.

To do that, the sustainability commission turned to the use of dashboards as a way of communicating with members of the public and soliciting their feedback. Such dashboards have great power in conveying information to citizens about a city's sustainability performance because they summarize information in an easy-to-read graphic form [30,31], but they should not be used uncritically because they can oversimplify complex situations and be manipulated by vested interests in some cases [32–36]. Figure 3 shows a preliminary dashboard used by the commission.

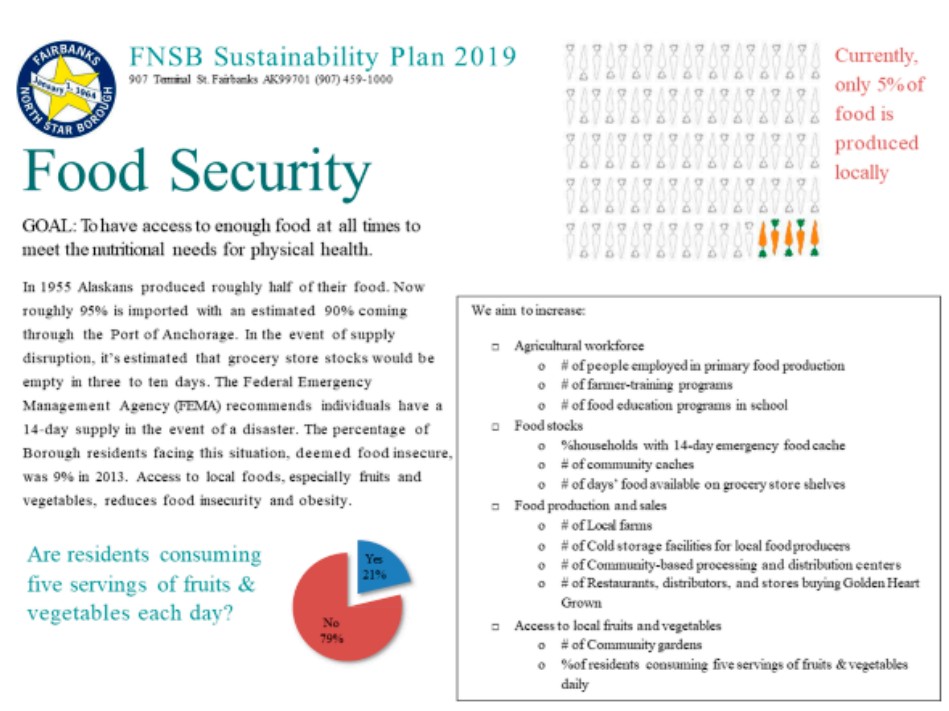

**Figure 3.** FNSB Dashboard on Food Security.

When the team of policymakers and academics presented the first draft of the dashboards to a meeting with the public on 5 March 2019, they received extensive feedback. The advice included making the dashboards less busy so that readers could get the main information in a glance, having schoolchildren contribute pictures and other ideas to the dashboards as a way of involving them in the process and attracting their parents and other community members as well, adding resources such as community gardens to the dashboards so that people would know where they can go to take action on

achieving the community's food security goals, ensuring that the dashboards send a positive message that it is possible to do something to make progress (for example, celebrating small victories) rather than overwhelming people with information that suggests that there is so much work to be done it is pointless to even try, identifying effective social science techniques that can nudge people forward in making choices that are useful for society, setting concrete goals, developing ways to measure the use of wild foods, and finding ways to show how the food, energy, and water systems are connected (e.g., reducing food waste would also help reduce the demand on water resources and be more energy efficient). The next steps for the project include incorporating this input to revise the dashboard and further publicize it on-line and through face-to-face meetings.

*3.6. Addressing Food Security in the FNSB*

The process of developing and implementing the sustainability plan shows that there is a group of activists interested in working with policymakers to increase food security in the area. For the public, there is a general sense that the borough needs to decrease reliance on outside, and vulnerable, supplies of food while increasing local production. There is no consensus on how to achieve this goal, however. The general public sees this as a problem that requires behavioral change that likely will take at least one generation to implement.

Policymakers receive extensive input from the public and a wealth of ideas. The main problem for them is to sift through the various suggestions that constituents provide them and identify the ones that are most likely to be effective. Collating the input from the public is where working with university-based experts and others can be most productive for the policymakers since they often do not have the time to read through all the material that public meetings generate and then distill it down to action points.

## 4. Discussion

What does this Fairbanks case study tell us about the questions that we raised in the discussion of the existing literature above? In broad terms, we found supportive evidence for the hypothesis that data could serve as a basis for bringing together policymakers, experts, and members of the public.

First, the Fairbanks case study shows one model of effective collaboration between universities and policymakers. The academics were able to create a first draft of the issues which could spur focused action by the policymakers. By interviewing stakeholders, they were able to help identify and articulate the key questions facing the community in the area of food security. Then, the policymakers on the sustainability commission were able to take the list of interests as a basis for soliciting structured feedback from community members to develop a sustainability plan that, after being approved by the sustainability commission, could guide community efforts to address the issues raised. The next step for the Fairbanks community will be working to ensure that the plan is implemented. As already demonstrated by the first efforts with the dashboards, the implementation phase will require extensive input from the public to define the key goals and metrics for measuring progress to achieve those goals.

Second, the outreach efforts demonstrated to borough officials that the community had wide interest in the issues the sustainability commission chose to focus on and were willing to participate in discussions of these questions to reach solutions. While the commission could not accommodate every suggestion, it sought to take into account the public input. This input is visible in the correlation between the comments gathered, published as an appendix to the plan, and the final set of indicators included in the plan.

Finally, it is worthwhile asking whether the final plan does a good job of addressing the needs of all the different elements in society in Fairbanks. There was a widespread perception in Fairbanks that the city had only a three-day supply of food, which meant that there would not be much available in the case of a disaster that cut the city off from outside supplies. It was not clear how accurate this number was: Did the three days only apply to local grocery and other food stores? Did it take into account stockpiles people had in their homes? Did it cover the kind of food that could be obtained from hunting

and fishing in the area, practices that are common in Alaska? Did it take into account most effectively the local food system and include elements that had proven to be important in other disasters, such as when Hurricane Maria cut off supplies to Puerto Rico? Infrastructure that proved relevant in Puerto Rico included food wholesale distributors, the catering abilities of the local airport, kitchens in local schools and hospitals, and restaurants spread throughout the island [37]. When these other resources are included, it is possible that Fairbanks has more food on hand than is commonly perceived.

Even as the problem the FNSB would face if a disaster were to cut off its food supply remains hard to quantify, there has been little planning to meet the needs of the various communities in the Borough. At the end of 2017, just over 4000 people received aid from the Supplement Nutritional Assistance Program (formerly known as food stamps) in the Borough, about 4 percent of the population [38]. The overall plan would serve the needs of this group by creating jobs, making food more available in the borough in general, and making locally grown fruits and vegetables more available. There will have to be more effort on delivering this food to under-served communities to ensure that low-income citizens without access to cars will be able to obtain the nutrition that they need. Typically, people dependent on this type of food aid do not have the resources to participate in the public policy-making process and as a result their voices are not frequently heard.

Other populations will also require special investigation, including the military bases in the borough, the university community, and indigenous populations. Fully understanding the needs of those communities will require additional research.

## 5. Conclusions

### 5.1. Data as a Means for Spurring Collaboration in the FNSB

This case study of the Fairbanks North Star Borough has sought to show how policymakers, academics, and members of the general public can collaborate in producing policies aimed at reducing community vulnerability to being cut off from its food supply. This community-defined goal is crucial at a time when it is likely that nearly all of the food consumed in the FNSB comes from out of state. The case study showed that data provides a basis around which the various stakeholders can collaborate.

The case study employed several methods to produce evidence that data can serve as a focal point for collaboration. First, the authors of this article were participant-observers of the overall process, of drawing up the FNSB sustainability plan and helping to begin the process of implementing it, and helped gather input from stakeholders that could be used as the basis for drafting the plan. They did this by interviewing key stakeholders and boiling down their responses to useful information for the policymakers. The university-based team also helped develop dashboards to communicate with the broader public and solicit feedback to identify key goals for the plan and metrics to measure progress toward achieving them.

The case study also provided a process tracing of the establishment of the FNSB sustainability commission and the adoption of the initial version of the sustainability plan. The detailed description of these events shows how the focus on collecting data from the community served as a basis for collaboration among the key stakeholders. The 21 interviews were crucial in moving things forward.

While the collaboration between universities and policymakers has led to the adoption of the FNSB sustainability plan, there is still considerable work to be done to be sure that the plan is implemented effectively and that it has a positive and measurable impact on food security in the Fairbanks area. To do this, members of the sustainability commission will have to convince the mayor and members of the Assembly to include sustainability plan priorities in future policies and the budget.

Equally important will be keeping the population engaged in the process. Here is where the collaboration between university and policymakers can continue, particularly in designing clear dashboards to communicate the broad goals of the sustainability plan and engage individual residents

in discussions of what the specific targets should be for increasing the food sector work force and expanding local food production. Hopefully future publications will be able to explore this process.

*5.2. Implications Beyond Fairbanks*

This case study has implications for other northern cities and urban areas more generally. The focus on data that underlay the ability of the various stakeholders to collaborate with each other is a universal approach that can be implemented in a wide variety of cases. Such a process of collecting input from experts, stakeholders, and the public can facilitate efforts to identify policy solutions to food security problems which can then be tested out to see how well they perform in reality.

What deserves further research is how to ensure just outcomes. Not only should the solutions proffered reduce the overall level of the FNSB's vulnerability to supply cut-offs, but they should also ensure that the most underserved communities also receive support. The deliberative processes discussed here go some of the way toward that end, but ultimately policymakers will need to ensure just outcomes in conditions where the most needy members of the community do not have the ability to participate in decision-making processes directly. Meeting that challenge will be a key test for the FNSB sustainability commission and its sustainability plan going forward.

**Author Contributions:** Conceptualization, R.W.O. and J.P.; methodology, R.W.O.; investigation, J.F., R.W.O., J.P., C.F.; data curation, J.F., R.W.O., J.P.; writing—original draft preparation, R.W.O., C.F.; writing—review and editing, R.W.O., J.F., J.P., C.F.; project administration, J.F., R.W.O., J.P., C.F.; funding acquisition, R.W.O.

**Funding:** This research was funded by the National Science Foundation, grant number 1545913.

**Acknowledgments:** We are grateful for the support from FNSB Sustainability Commission and the 21 interviewees.

**Conflicts of Interest:** The authors were heavily involved in the processes that they describe here. J.F. is a member of the FNSB sustainability commission and played a central role in the development of its sustainability plan. R.W.O. and J.P. worked as unpaid consultants for the commission and provided support in the development of the sustainability plan by interviewing stakeholders and drawing up a list of their primary concerns. C.F. helped developed the dashboards used by the commission. The participant–observer nature of the work gave the authors insight into the policy-making process and they used their insider knowledge to describe that process. The funders had no role in the design of the study; in the collection, analyses, or interpretation of data; in the writing of the manuscript, or in the decision to publish the results.

## Appendix A. Survey Instrument

Semi-Structured Interview Questions for FNSB Food Security Stakeholders

1. What do you consider the most important issues in Fairbanks?
2. What do you see as the main problems in the area of food for Fairbanks?
3. Why are these particular problems more important than other problems?
4. Do you see any particular solutions to these problems? Is anything being done along these lines?
5. Do you have data that could be used to measure progress toward these solutions?
6. Who (individuals, groups, public agencies) would be most appropriate in carrying out these solutions?
7. What kinds of resources are needed to address these problems? Money, coordination, citizen participation, other resources?
8. Is the city/borough making plans to address these problems?
9. What is the role of business in addressing these problems? Will business-oriented solutions lead to economic growth in this area? Are these small businesses? Is there a strong environment to support the growth of small businesses? Where are most small businesses focused now?
10. What prevents these problems from being addressed? Obstacles?
11. How can Fairbanks develop new local talent to address these problems? What is the role of young people here? Is the education system giving people the skills that they need?

12. Does the solution require resources from outside Fairbanks – expertise, capital, other? Can this come from the rest of the USA or will it be international? Alternatively, are Fairbanks companies thinking of exporting their expertise to other parts of the U.S. or the rest of the world?

13. Did we leave anything out?

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
