# Peer review of "Strengthening Food Security Near the Arctic Circle: Case Study of Fairbanks North Star Borough, Alaska"

_sustainability, doi:10.3390/su11102722_

Round 1

Reviewer 1 Report

I found this paper interesting, but the methodology is not described, only details of Faibanks. I suggest that explain de process of work with the participants and how they gave their proposals. 

It is important to detail the instrument that was used to develop the participation of Fairbanks in this study.

In results, I suggest separate the report of the group analysis and the Food Security report.

How do participants resolved by them-selves the problem of food security (their perspective)?

How do policy-makers obtain the information for improve policies?

Do you find some phenomenon that could be reproduced in another context?

Author Response

I found this paper interesting, but the methodology is not described, only details of Faibanks. I suggest that explain de process of work with the participants and how they gave their proposals. 

We greatly revised the methodology section to explain our process in much greater detail.

It is important to detail the instrument that was used to develop the participation of Fairbanks in this study.

We described the instrument in the body of the text and added the questionnaire as Appendix A.

In results, I suggest separate the report of the group analysis and the Food Security report.

We added section 3.6 to address the report.

How do participants resolved by them-selves the problem of food security (their perspective)?

We added this discussion to section 3.6. The general public sees the need for behavioral change, but recognizes that this will likely take at least one generation.

How do policy-makers obtain the information for improve policies?

Ideas from the public addressed with experts (section 3.6)

Do you find some phenomenon that could be reproduced in another context?

Yes! We added a final section of the paper that explains this.

Reviewer 2 Report

It has been my pleasure to review “Strengthening Food Security at the Arctic Circle: Case Study of Fairbanks Alaska” Manuscript ID: sustainability-458047. The manuscript could be a good fit for ‘Sustainable Urban and Rural Development’

Section. However, the current formulation is not convincing; I have noted some room for improvement below: 

In the current formulation the paper reads a lot like a project report without a clear linked to scientific theory, approaches, methods etc.

- One major issue is that the Materials and Methods fail to outline what was done.

- Another very major issue is that it is not clear what the authors see as the benefit of the research article. In the abstract the authors claim that the “analysis pushes forward the boundaries of science in food security and community-engagement studies by examining how university-based teams and policy makers can work together to improve food security.” After reading the manuscript I cannot see how this grand vision was met.

- At the end of the Introduction the authors go on to claim that the work seeks to provide analysis of responses of policy makers to academics and then lays out ‘additional research’.  In my view this spot in the manuscript (the end of the introduction) should outline something akin to objectives, aims and hypotheses. Ideally after reading this paragraph we would have a sense for why the study was done.

- The methods are not only very short and fail to indicate the approaches and tools, they also indicate some potential bias in the authorship (not just shown here but also elsewhere in the manuscript), which should be stated in the conflict of interest.

Minor issues:

Introduction

Rather many parts that are off topic. It is not clear how data on the international trends in obesity, hunger, etc. are related to the study in Alaska.

36 It seems it would be better to replace ‘it is necessary’ with ‘this will be necessary in order to’

41 seems off topic

46 The phrase ‘advent of modern society’ seems an awkward way to introduce the idea of industrial ag. and international trade. Better to give a date or an era.

47 The term ‘entirely’ is probably too extreme

55 The term ‘food’ seems too unspecific, does it that what is mentioned in the 2018 paper?

57-58 It seems it would be better to remove ‘for in state consumption’

65-66 It comes across as off-topic to mention Fairbanks and food security ‘around the world’. Two seemingly unrelated ideas.

73 Example of the inconsistent use of terms referring to the continental US. Here is ‘lower 48 U.S. states’ elsewhere it is called differently

74 ‘given way to more lucrative subdivisions to house’ should be ‘disappeared to make room for’ (frankly this language comes across as highly subjective / biased, the authors are clearly not impressed with this change).

77 remove ‘culture to’

77 ‘for’ after ‘support’

79 remove ‘million’ after $400

91 citation missing at end of line

92-96 different than abstract (no real direct impact)

Materials and Methods

Lines 106-110 are discussion (starting with ‘While”) – In Discussion (when moved) – after ‘While’ add ‘we believe’ and after Fairbanks add ‘as a case study’.

106 ‘significant’ often has a very specific meaning in science. - Probably better to use another term when not referring to a test of some kind.

Line 111-121 much of this text outlines the role that the authors pled in the establishment of the program, which is being assessed and belongs in the ‘conflicts of Interest’.

What is missing:

Which interview methods were followed?

116 semi-structured in which way? Citation would be nice here. What questions were asked? I suggest a supplementary or annex.

110 if ‘numerous dimensions’ are important for the study and framed your methods then explain what they are and how they framed the work that is presented in the results

114-115 21 individual interviews with 21 stakeholders?

Who were the stakeholders?

117-118 explain ‘visual dashboards’ in detail, citations (how were these used to achieve the results?)

118 Now we are getting the information that the ‘Sustainability plan’ has been drafted and ‘will be’ communicated. So this study happens in the space between the time when the ‘Sustainability plan’ is completed and the ‘relevant stakeholders’ find out about it.

Why were the ‘relevant stakeholders’ not involved in drafting the plan?

119 the study ‘employs a process-tracing approach’ (detailed explanation and citations missing) how does this approach lead to the results presented?

119-120 is ‘track the process’ of the commission in developing the ‘Sustainability plan’ the aim of this work? If so, are there any suggestions for this process, could it be that given the authorship the process will receive a bad review through the methods applied (seems doubtful in the current formulation and clear bias in the authorship)?

Results

The Results read as a chronology of the establishment of the ‘Sustainability Commission‘. Frankly, I cannot really get through the results or make much sense of them without a clear raison d'être for the paper.

125-127 this text is for the ‘Introduction’ section

127 ‘as we show’ is about the results of the analysis tools described in methods? Then it is ‘our results show’

127 remove ‘however’

163-178 this text is for the ‘Introduction’ section

164 more conflict of interest

Table 1. Could be supported by some values for weighting of the relative risk according to stakeholders. Both qualitative (degree to which they think it is an issue) and quantitative (percent response)

193 ‘stake holders’ should be ‘stakeholders (throughout manuscript)

Figure 1 appears to be a screenshot of a video. Would be nice to have a less blurry and an actual photo. Have al participants agreed to be photographed and included here?

Discussion

Watch out for confusing reference to the work at hand (probably part of the confusion about the over-arching purpose of the study). The ‘Fairbanks case study’ and ‘Sustainablity Commission’ seem to be used interchangeably.

357 ‘allows us to draw conclusions’ bit it is not clear which

372 maybe use this as theoretical option for tying together the rest of the manuscript? Could be a theory

375 This para. is introduction

387-388 example of how this comes across as a project report.

Link to Puerto Rico is rather topical.

Conclusions

Are about the ‘case study’ only and should be broadened. The conclusions argue mainly that the FNSB be continued and that the government should support it. What about relevance to wider world?

The conclusions should offer a synopsis of the main parts of the discussion (essentially a condensed version) based on a strong methods descriptions, and results that follow in detail.

Author Response

It has been my pleasure to review “Strengthening Food Security at the Arctic Circle: Case Study of Fairbanks Alaska” Manuscript ID: sustainability-458047. The manuscript could be a good fit for ‘Sustainable Urban and Rural Development’

Section. However, the current formulation is not convincing; I have noted some room for improvement below: 

In the current formulation the paper reads a lot like a project report without a clear linked to scientific theory, approaches, methods etc.

- One major issue is that the Materials and Methods fail to outline what was done.

We removed the discussion of Fairbanks from this section and greatly increased the discussion of what we did.

- Another very major issue is that it is not clear what the authors see as the benefit of the research article. In the abstract the authors claim that the “analysis pushes forward the boundaries of science in food security and community-engagement studies by examining how university-based teams and policy makers can work together to improve food security.” After reading the manuscript I cannot see how this grand vision was met.

We revised these grand claims to a more focused assertion that “This analysis makes a contribution by examining how university-based teams and policy makers can work together to improve food security. Specifically, it examines the most profitable areas for collaboration for academics and policy-makers and how they can best incorporate public input into policies and research addressing these policies.”

- At the end of the Introduction the authors go on to claim that the work seeks to provide analysis of responses of policy makers to academics and then lays out ‘additional research’.  In my view this spot in the manuscript (the end of the introduction) should outline something akin to objectives, aims and hypotheses. Ideally after reading this paragraph we would have a sense for why the study was done.

Revised the introduction to lay out the objectives.

- The methods are not only very short and fail to indicate the approaches and tools, they also indicate some potential bias in the authorship (not just shown here but also elsewhere in the manuscript), which should be stated in the conflict of interest.

 We revised the conflict of interest statement.

Minor issues:

Introduction

Rather many parts that are off topic. It is not clear how data on the international trends in obesity, hunger, etc. are related to the study in Alaska.

Deleted this general overview of the food system.

36 It seems it would be better to replace ‘it is necessary’ with ‘this will be necessary in order to’

deleted

41 seems off topic

deleted

46 The phrase ‘advent of modern society’ seems an awkward way to introduce the idea of industrial ag. and international trade. Better to give a date or an era.

This discussion has been expanded to show that there is doubt that humans ever were able to live sustainably in the interior of Alaska

47 The term ‘entirely’ is probably too extreme

deleted

55 The term ‘food’ seems too unspecific, does it that what is mentioned in the 2018 paper?

Yes the 2018 report examined the food supply

57-58 It seems it would be better to remove ‘for in state consumption’

done

65-66 It comes across as off-topic to mention Fairbanks and food security ‘around the world’. Two seemingly unrelated ideas.

Focused on Fairbanks

73 Example of the inconsistent use of terms referring to the continental US. Here is ‘lower 48 U.S. states’ elsewhere it is called differently

Made consistent as continental U.S.

74 ‘given way to more lucrative subdivisions to house’ should be ‘disappeared to make room for’ (frankly this language comes across as highly subjective / biased, the authors are clearly not impressed with this change).

Replaced this wording with more neutral phrase

77 remove ‘culture to’

done

77 ‘for’ after ‘support’

added

79 remove ‘million’ after $400

done

91 citation missing at end of line

Revised this section

92-96 different than abstract (no real direct impact)

Revised and ensured conformance with abstract

Materials and Methods

Lines 106-110 are discussion (starting with ‘While”) – In Discussion (when moved) – after ‘While’ add ‘we believe’ and after Fairbanks add ‘as a case study’.

Moved this paragraph to the Introduction and added the suggested text.

106 ‘significant’ often has a very specific meaning in science. - Probably better to use another term when not referring to a test of some kind.

Replaced with high

Line 111-121 much of this text outlines the role that the authors pled in the establishment of the program, which is being assessed and belongs in the ‘conflicts of Interest’.

 Added to the conflicts of interest

What is missing:

Which interview methods were followed?

Citation added

116 semi-structured in which way? Citation would be nice here. What questions were asked? I suggest a supplementary or annex.

Citation and appendix of questions asked has been added

110 if ‘numerous dimensions’ are important for the study and framed your methods then explain what they are and how they framed the work that is presented in the results

Deleted this ill-advised phrase

114-115 21 individual interviews with 21 stakeholders?

Who were the stakeholders?

Description added

117-118 explain ‘visual dashboards’ in detail, citations (how were these used to achieve the results?)

Added a more extensive discussion of the dashboards in section 3.5, including citations on dashboards.

118 Now we are getting the information that the ‘Sustainability plan’ has been drafted and ‘will be’ communicated. So this study happens in the space between the time when the ‘Sustainability plan’ is completed and the ‘relevant stakeholders’ find out about it.

Why were the ‘relevant stakeholders’ not involved in drafting the plan?

Sorry for the confusion here. Stakeholders were involved in writing the plan. The purpose of the dashboards is to communicate with the broader public about the main focuses of the plan and to solicit specific suggestions from the public on setting specific goals and identifying data and metrics to measure progress toward achieving these goals.

119 the study ‘employs a process-tracing approach’ (detailed explanation and citations missing) how does this approach lead to the results presented?

Added more details and references on process tracing. The main gist of the approach in this case is to show how the process of including university experts and the general public improved the output of food security in the Fairbanks North Star Borough.

119-120 is ‘track the process’ of the commission in developing the ‘Sustainability plan’ the aim of this work? If so, are there any suggestions for this process, could it be that given the authorship the process will receive a bad review through the methods applied (seems doubtful in the current formulation and clear bias in the authorship)?

 We added recommendations on the process to the conclusion.

Results

The Results read as a chronology of the establishment of the ‘Sustainability Commission‘. Frankly, I cannot really get through the results or make much sense of them without a clear raison d'être for the paper.

We added material above to better explain the purpose of the paper.

125-127 this text is for the ‘Introduction’ section

We moved some of the discussion to the methods section, but left here the discussion of the creation of the Sustainability Commission because it is crucial to our discussion of the evolution of the process.

127 ‘as we show’ is about the results of the analysis tools described in methods? Then it is ‘our results show’

done

127 remove ‘however’

done

163-178 this text is for the ‘Introduction’ section

We moved some of the discussion to the methods section, but left here the discussion of the creation of the Sustainability Commission because it is crucial to our discussion of the evolution of the process.

164 more conflict of interest

 Noted

Table 1. Could be supported by some values for weighting of the relative risk according to stakeholders. Both qualitative (degree to which they think it is an issue) and quantitative (percent response)

We intended this Table as simply a list of the issues raised.

193 ‘stake holders’ should be ‘stakeholders (throughout manuscript)

done

Figure 1 appears to be a screenshot of a video. Would be nice to have a less blurry and an actual photo. Have al participants agreed to be photographed and included here?

deleted

Discussion

Watch out for confusing reference to the work at hand (probably part of the confusion about the over-arching purpose of the study). The ‘Fairbanks case study’ and ‘Sustainablity Commission’ seem to be used interchangeably.

 This section has been extensively rewritten.

357 ‘allows us to draw conclusions’ bit it is not clear which

Section clarified

372 maybe use this as theoretical option for tying together the rest of the manuscript? Could be a theory

Yes, added to main hypothesis

375 This para. is introduction

Moved to lit review section

387-388 example of how this comes across as a project report.

Link to Puerto Rico is rather topical.

Conclusions

Are about the ‘case study’ only and should be broadened. The conclusions argue mainly that the FNSB be continued and that the government should support it. What about relevance to wider world?

 The final section now address  implications beyond Fairbanks

The conclusions should offer a synopsis of the main parts of the discussion (essentially a condensed version) based on a strong methods descriptions, and results that follow in detail.

The conclusion has been updated to include this discussion.

Reviewer 3 Report

This paper reports on the nitty gritty of food security work in an interesting, remote region. I think the paper shows the nuts and bolts about how community-academia partnerships can work, specifically working toward building food security. This is a very interesting aspect of the manuscript, more than the details about Alaska. That being said, I think the way it is presented is good. The reader can read between the lines to get to the broader significance. Another way of saying it is that this kind of community work is not sexy, but super important. The authors may consider adding reference to the following work that was previously done in Fairbanks: 

Meadow, Alison. 2012. “Assessing Access to Local Food System Initiatives in Fairbanks, Alaska.” Journal of Agriculture, Food Systems, and Community Development 2 (2): 217–36. https://doi.org/10.5304/jafscd.2012.022.006.

Meadow, Alison. 2013. “Alternative Food Systems at Ground Level: The Fairbanks Community Garden.” Journal of Ecological Anthropology 16 (1): 76–84. http://dx.doi.org/10.5038/2162-4593.16.1.6.

and this one on community work, a designation within which I think this project falls: 

Loring, Philip A., S. Craig Gerlach, and Henry J.F. Penn. 2016. “‘Community Work’ in a Climate of Adaptation: Responding to Change in Rural Alaska.” Human Ecology 44 (1): 119–28. https://doi.org/10.1007/s10745-015-9800-y.

None of these have to be cited for the paper to be publishable but the authors may find them helpful.